

# Genome-wide analysis of long non-coding RNA expression profile in porcine circovirus 2-infected intestinal porcine epithelial cell line by RNA sequencing

Manxin Fang, Yi Yang, Naidong Wang, Aibing Wang, Yanfeng He, Jiaoshun Wang, You Jiang and Zhibang Deng

Hunan Agricultural University, Hunan Provincial Key Laboratory of Protein Engineering in Animal Vaccines, Changsha, Hunan, China

## ABSTRACT

Porcine circovirus-associated disease (PCVAD), which is induced by porcine circovirus type 2 (PCV2), is responsible for severe economic losses. Recently, the role of noncoding RNAs, and in particular microRNAs, in PCV2 infection has received great attention. However, the role of long noncoding RNA (lncRNA) in PCV2 infection is unclear. Here, for the first time, we describe the expression profiles of lncRNAs in an intestinal porcine epithelial cell line (IPEC-J2) after PCV2 infection, and analyze the features of differently expressed lncRNAs and their potential target genes. After strict filtering of approximately 150 million reads, we identified 13,520 lncRNAs, including 199 lncRNAs that were differentially expressed in non-infected and PCV2-infected cells. Furthermore, *trans* analysis found lncRNA-regulated target genes enriched for specific Gene Ontology terms ($P < 0.05$), such as DNA binding, RNA binding, and transcription factor activity, which are closely associated with PCV2 infection. In addition, we analyzed the predicted target genes of differentially expressed lncRNAs, including *SOD2*, *TNFAIP3*, and *ARG1*, all of which are involved in infectious diseases. Our study identifies many candidate lncRNAs involved in PCV2 infection and provides new insight into the mechanisms underlying the pathogenesis of PCVAD.

## INTRODUCTION

Infectious diseases threaten pig production, which is an important source of meat (*Karuppannan & Opriessnig, 2017*). Porcine circovirus type 2 (PCV2) in the family Circoviridae is one of the most important pathogens affecting the pig population (*Todd et al., 1991*). This small, non-enveloped, and circular DNA virus is the causative agent of porcine circovirus-associated disease (PCVAD), which can manifest as PCV2-systemic disease, porcine dermatitis, nephropathy syndrome, porcine respiratory disease complex, PCV2-enteric disease, reproductive failure, and acute pulmonary edema (*Meng, 2013*; *Segales, 2012*; *Segales, Allan & Domingo, 2005*). Increasing evidence

Corresponding author
Zhibang Deng,
zbangd@hunau.edu.cn

indicates that PCVAD is a disease that causes considerable economic damage (*Lekcharoensuk et al., 2004*; *Segales, 2012*).

Recent studies showed that noncoding RNAs play important regulatory roles in PCVAD (*Hong et al., 2015*; *Wang et al., 2017*). However, studies of PCV2-induced PCVAD have mainly focused on microRNA rather than on long non-coding RNA (lncRNA). The lncRNAs are defined as non-protein-coding transcripts greater than 200 nucleotides in length (*Batista & Chang, 2013*). They play significant roles in cellular activities, such as genome regulation, and cell growth, differentiation, and apoptosis (*Batista & Chang, 2013*; *Yang et al., 2013*). Furthermore, many viruses, such as influenza virus, enterovirus, and porcine reproductive and respiratory syndrome virus (PRRSV), can alter the expression of lncRNAs (*Winterling et al., 2014*; *Yin et al., 2013*; *Zeng et al., 2018*). Dysregulation of lncRNAs may also lead to diseases in pigs (*Gao et al., 2017*; *Zhou et al., 2014*).

No studies have examined the role of swine lncRNAs in PCV2-associated PCVAD. Enteritis is a common clinical manifestation of PCV2 infection because the intestinal mucosa is the initial site of PCV2 infection (*Chae, 2005*; *Kim et al., 2004*). The intestinal porcine epithelial cell line IPEC-J2 is a non-transformed columnar epithelial cell line that was isolated from the neonatal piglet mid-jejunum by Helen Berschneider et al. (*Orlando et al., 1989*). IPEC-J2 is a well-validated model for studying the processes involved in pathogenic infections in the porcine intestinal epithelium (*Arce et al., 2010*; *Koh et al., 2008*; *Skjolaas et al., 2007*). Thus, in the current study, we evaluated the expression profiles of lncRNAs in an intestinal porcine epithelial cell line (IPEC-J2) after PCV2 infection by RNA sequencing and validated the results by quantitative real-time polymerase chain reaction (qRT-PCR). Moreover, Gene Ontology (GO) and pathway analyses were conducted to identify the biological roles of lncRNAs that were differentially expressed after PCV2 infection. To our knowledge, this is the first study to describe the aberrant lncRNA expression profile in response to PCV2 infection in IPEC-J2.

## MATERIALS AND METHODS

### Cell culture and virus

IPEC-J2 cells lines free from porcine circovirus (Guangzhou Jennio Biotech Co., Ltd., Guangzhou, China) were used in the present study. Cells were cultured in DMEM-F12 medium supplemented with 5% FBS. All cells were maintained at 37 °C in a humidified incubator containing 5% $CO_2$. Here, IPEC-J2 cells were cultured on six-well plastic tissue culture plates (Corning, Inc., Corning, NY, USA) at a density of $3 \times 10^5$/well. The virus PCV2b (GenBank accession number: KJ867555) used in this work was provided by Hunan Provincial Key Laboratory of Protein Engineering in Animal Vaccines and stored at −80 °C. The infectious titer of the PCV2 virus prepared from IPEC-J2 cells was $10^{4.7}$ $TCID_{50}$/ml.

### Virus infection

Porcine circovirus type 2 infection was performed as previously described (*Yan, Zhu & Yang, 2014*). IPEC-J2 cells were grown to approximately 85% confluence and washed twice with phosphate-buffered saline. Next, cells were infected with PCV2 at

**Table 1 Quality of total RNA.**

| Sample name | Conc. (μg/μL) | Total (μg) | 260/280 | RIN | 28S/18S | QC evaluation |
|---|---|---|---|---|---|---|
| IPEC-A1-A | 2.04 | 61.32 | 2.06 | 10 | 2.1 | A |
| IPEC-A2-A | 1.98 | 59.31 | 2.08 | 10 | 2.0 | A |
| IPEC-A3-A | 2.03 | 60.82 | 2.06 | 10 | 2.0 | A |
| IPEC-B1-B | 1.96 | 58.93 | 2.08 | 10 | 1.9 | A |
| IPEC-B2-B | 1.85 | 55.62 | 2.08 | 10 | 2.0 | A |
| IPEC-B3-B | 2.22 | 66.48 | 2.03 | 10 | 1.9 | A |

$3 \times 10^{2.5}$ TCID$_{50}$/ml. After 1 h of adsorption, infected cells were cultured in fresh medium supplemented with 2% FBS. Uninfected cells were used as a negative control. Both the PCV2-infected and uninfected cells were harvested 36 h post-infection and used for total RNA extraction.

## Library preparation and Illumina sequencing

Total RNA was extracted from each cell group using Trizol (Invitrogen, Carlsbad, CA, USA). The RNA amount and purity of each sample was quantified by NanoDrop ND-1000 (NanoDrop, Wilmington, DE, USA) (Table 1). Moreover, the RNA integrity was assessed by Agilent 2100 Bioanalyzer (Agilent, Santa Clara, CA, USA) (Table 1). In this study, RNA with a RIN (RNA integrity number) = 10 was used for library preparation. After reverse transcription, the purified first-strand cDNA was subjected to PCR amplification. The preparation for library and sequencing were performed according to previous studies (*Wang et al., 2016*) by LC-Bio (Hangzhou, China). Finally, RNA sequencing was assessed by the HiSeq 4000 (Illumina, San Diego, CA, USA) on a 150 bp paired-end run.

## Quality control

First, raw reads in the FASTA format were processed through in-house Perl scripts. During this step, clean reads (clean data) were obtained by discarding reads either containing adapter or over 10% poly-N sequences, and low quality reads (>50% of bases whose Phred scores were <5%). Meanwhile, Q20, Q30, and GC content were calculated for the clean data. All downstream analyses were carried out based on clean reads with high quality.

## Mapping to the reference genome

Clean reads were aligned to the porcine reference genome (Sus scrofa 10.2) downloaded from the Genome website using TopHat v2.0.9 run with default parameters (*Trapnell et al., 2012*).

## Transcriptome assembly

The mapped reads of each cell group were assembled using both StringTie (v1.3.3) and Cufflinks (v2.1.1) based on default parameters as previously described (*Pertea et al., 2015*; *Trapnell et al., 2010*).

## Identification of lncRNAs

In addition to lncRNAs identified by BLAST searches, we also predicted novel lncRNAs from clean reads (Fig. 1). Novel putative lncRNAs had to meet three requirements that
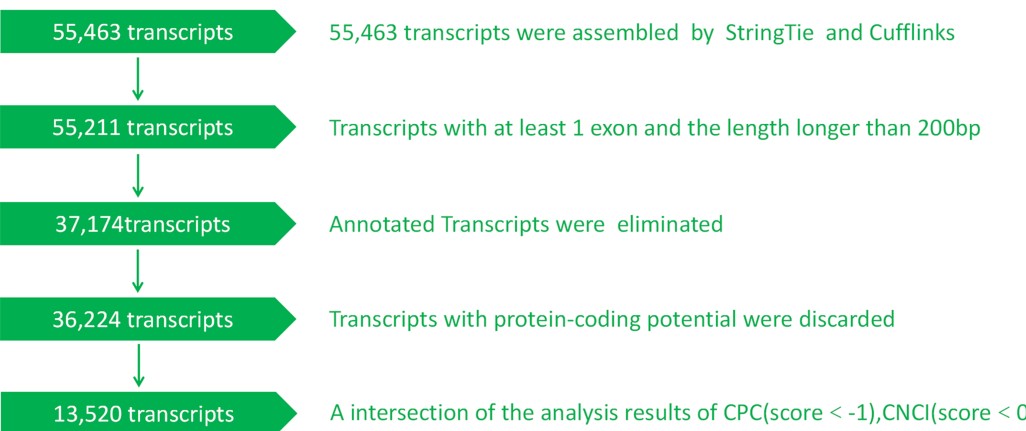

**Figure 1** **The developed pipeline for identifying putative lncRNAs.** Briefly, 55,463 transcripts were assembled and 13,520 novel lncRNAs were identified.   

have also been described previously: (1) transcripts should be longer than 200 bp and contain more than one exon; (2) the fragments per kilobase of transcript per million mapped reads (FPKM) score should be more than 0.5; 3) transcripts should have no coding potential. (*Ran et al., 2016*; *Weng et al., 2017*). In addition, the FRKM was transformed by the following formula:

$$FPKM = \frac{\text{Total exon Fragments}}{\text{Mapped reads (millions)} \times \text{exon length (kb)}}$$

To distinguish protein-coding and non-coding sequences effectively, Coding-Non-Coding-Index (CNCI v2) profiles were analyzed independent of known annotations according to the following parameters: length ≥200 bp and exon ≥1 and score ≤0 (*Meng, 2013*; *Segales, Allan & Domingo, 2005*; *Sun et al., 2013*; *Todd et al., 1991*). The Coding Potential Calculator (CPC 0.9r2) mainly assesses the extent and quality of the open reading frame (ORF) in transcript sequences and compares it to sequences in a protein sequence database to differentiate coding and non-coding transcripts (*Kong et al., 2007*). All transcripts with CPC scores ≤−1 were removed. This filtering process yielded transcripts without coding potential that formed our candidate set of lncRNAs and were used for subsequent analysis.

## Distribution of lncRNAs along each chromosome

Based on the location with respect to protein-coding genes, lncRNAs are classified into three types: intergenic lncRNAs (lincRNAs), intronic lncRNAs, and anti-sense lncRNAs (*Harrow et al., 2012*; *St Laurent, Wahlestedt & Kapranov, 2015*). After comparing with known mRNAs via the class_code module in cuffcompare, the putative lncRNAs were separated into these three classes (*Yan, Zhu & Yang, 2014*). Next, all these kinds of lncRNAs were mapped to the porcine genome separately to determine their chromosomal distribution. Briefly, lncRNAs were aligned by short blast, followed by best hit analysis in short 500 kb segments. To evaluate their chromosomal distribution, we used the start sites of lncRNAs in the chromosomes counted in the pig reference genome (Susscrofa10.2).

## Target gene prediction

Transcripts without coding potential constituted our candidate set of lncRNAs. Next, coding genes were searched within a range of 10–100 kb upstream or downstream of each candidate lncRNA for the *cis* target gene (*Yan, Zhu & Yang, 2014*; *Zhang et al., 2017a*). For *trans* interactions, we determined the level gene expression based on Pearson's correlations. Briefly, the Pearson's correlation coefficients ($R$) between lncRNAs and mRNAs were calculated using the R statistical package, and lncRNA target genes were predicted with $R \geq 0.95$ (*Li et al., 2014*).

## Quantification of gene expression levels

To determine gene expression levels, we calculated the FPKMs for both lncRNAs and coding genes in each cell group using Cuffdiff (v2.1.1) software run with default parameters as previously described (*Trapnell et al., 2010*).

## Differential expression analysis

To determine differential expression in digital transcript or gene expression datasets, we utilized a model based on a negative binomial distribution provided by Cuffdiff software. In addition, transcripts or genes in biological replicates with $P < 0.05$ were classed as differentially expressed. For non-biological replicates, transcripts or genes with $P < 0.05$ and an absolute value of log2 (fold change) more than one were assigned as the threshold for significant differential expression. Coverage signals used to generate heatmaps were obtained using heatmaply (a R package) based on the *z*-score obtained following dimensionality reduction of FPKM values (*Galili et al., 2018*).

## Validation of RNA-Seq data by qRT-PCR

To further confirm the reliability of the RNA-seq data, qRT-PCR assays were performed. Briefly, single-stranded cDNA was generated using the RevertAid kit (Fermentas Life Science, Burlington, ON, Canada) with random primers. Real-time PCR was conducted using the SYBR Green q-PCR SuperMix (Bio-Rad, Hercules, CA, USA). The primers used are listed in Table 2. Subsequently, Ct values were acquired with manual thresholds using the 7500 System SDS software (ABI, Foster City, CA, USA). Levels of lncRNA expression were normalized to the level of GAPDH expression according to the $\Delta\Delta$CT method. The lncRNA expression levels between different groups were compared using $2^{-\Delta\Delta\mathrm{CT}}$. *P*-values $< 0.05$ were considered statistically significant.

## Gene Ontology and KEGG pathway enrichment analysis

Analysis of GO enrichment for differentially expressed lncRNAs-regulated target genes was performed using the GOseq R package as previously described (*Young et al., 2010*). For KEGG pathway enrichment of differentially expressed lncRNA-regulated target genes (*Young et al., 2010*), the KOBAS software program was used as described previously (*Mao et al., 2005*).
| Table 2 Primer list. | | | | |
|---|---|---|---|---|
| Gene | ID | | Primer sequence | Product length |
| MSTRG.19762.1 | | F | CGACGACAAAACGAGAGTCA | 196 |
| | | R | AATTCTTGAAAAGCGGCTGA | |
| MSTRG.1454.1 | | F | CACCTTCTCCATTGCTCCAT | 207 |
| | | R | CATGCTGCTTTATTGCCAAA | |
| MSTRG.641.1 | | F | TGCTCTCGGTCTCCCTTCTA | 202 |
| | | R | TTGGGATCCTCGACATTCTC | |
| MSTRG.31692.1 | | F | CGTGAGAGATGCCATTCAGA | 215 |
| | | R | AGGACTACCCTCCACCGAGT | |
| MSTRG.385.1 | | F | TCCGACTAGGAACCATGAGG | 173 |
| | | R | TCCCAGGCTAGGGGTCTAAT | |
| MSTRG.2965.1 | | F | CTCAGTGGGTTAAGGGTCCA | 161 |
| | | R | GTTTTCTGGCTGCACATACG | |
| MSTRG.15360.1 | | F | ATAAGGTTGCGGGTTCGAT | 172 |
| | | R | TCCCTCAGCATATGGAGGTT | |
| MSTRG.22503.1 | | F | AACCAACTCGGTTGTTCCTG | 239 |
| | | R | CCTATCGCCTTTCTCTGTGC | |
| MSTRG.5484.1 | | F | GAGCCGCATCTGCTACCTAC | 207 |
| | | R | ACACGGTTCCGGACTTAGTG | |
| GAPDH | 396823 | F | TCGGAGTGAACGGATTTGGC | 189 |
| | | R | TGACAAGCTTCCCGTTCTCC | |

# RESULTS

## Reads and mapping of RNA-seq in IPEC-J2 cells

Total RNAs without rRNA for six samples including mock (IPEC_A1-A, IPEC_A2-A, IPEC_A3-A) and PCV2-infected IPEC-J2 cells (IPEC_B1-B, IPEC_B2-B, IPEC_B3-B) were sequenced. After discarding low-quality and adapter sequences, approximately 100 million clean reads were obtained for each cell group. The percentage of clean reads was approximately 98% (Table S1). Next, clean reads were mapped to the pig reference genome (Susscrofa10.2) using TopHat; >69% of the clean reads were mapped (Table S2). More than 58% of reads were uniquely mapped, 42% were non-splice reads, and approximately 16% were splice reads (Table S2). Additionally, more than 60% of the reads were located on exons, whereas approximately 13% of the reads were on introns; the remaining reads were in intergenic regions (Figs. 2A–2F; Table S3).

## Assessing the quality of RNA-seq data

Fragments per kilobase of transcript per million mapped reads values were used to measure gene expression levels in IPEC-J2 cells. Based on the distribution profile for all transcripts shown in Fig. 3A, the patterns of expression between the control group (IPEC_A1-A, IPEC_A2-A, IPEC_A3-A) and PCV2 infection group (IPEC_B1-B, IPEC_B2-B, IPEC_B3-B) were similar. The FPKM density in the control and PCV2 infection groups was consistent (Fig. 3B). Importantly, the RNA-Seq Pearson
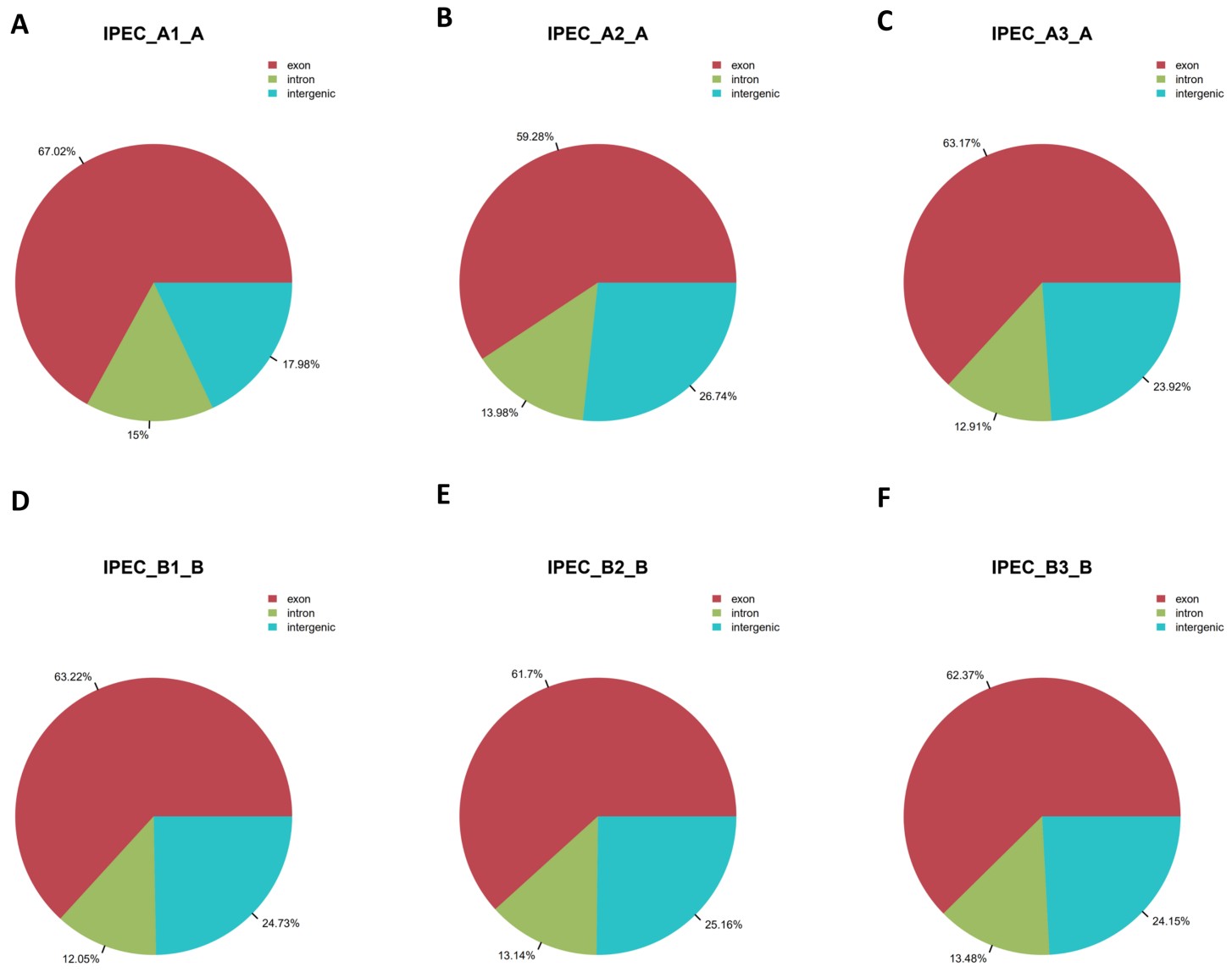

**Figure 2 Region distribution of raw reads.** The summary of the region distribution of raw reads in the genome from the control groups IPEC_A1-A (A), IPEC_A2-A (B), IPEC_A3-A (C), and PCV2 infection groups IPEC_B1-B (D), IPEC_B2-B (E), IPEC_B3-B (F).

correlation coefficients of transcript levels were greater than 0.84 in the control group and greater than 0.94 in the PCV2 infection group, indicating the rationality of the experimental design between these two groups and similarity of expression within the groups (Fig. 3C).

## Identification of lncRNAs in IPEC-J2 cells

Reads were assembled using StringTie and selected lncRNAs were spliced using Cufflinks. After these rigorous selections, 13,520 novel lncRNAs were identified, including 10,975 lincRNAs, 2,182 intronic lncRNAs, and 301 anti-sense lncRNAs (Fig. S1A; Table S4). To further evaluate whether these lncRNAs had coding potential, we predicted the

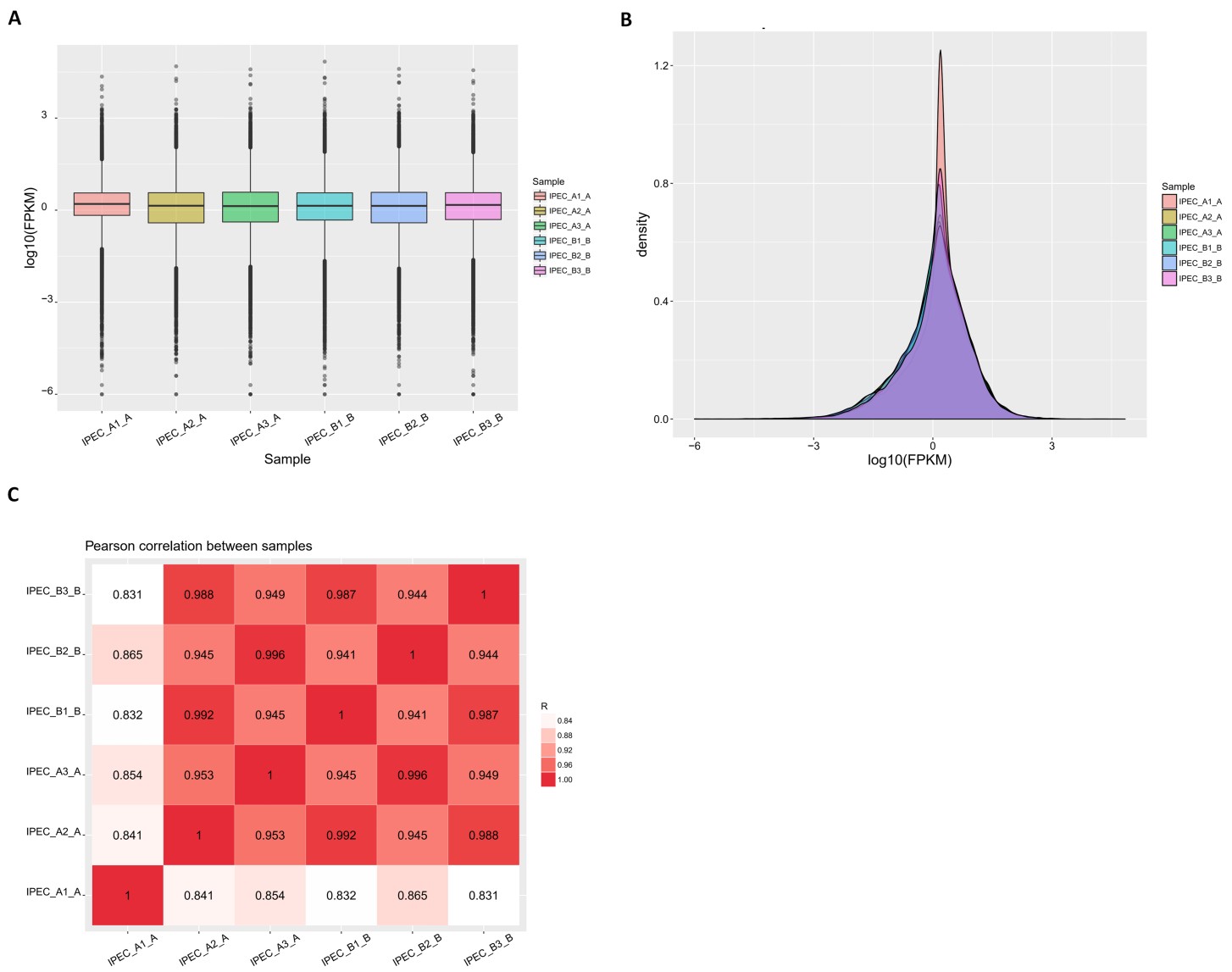

**Figure 3 Reads and mapping results of RNA deep sequencing.** (A) The FPKM distribution is shown as a box plot. (B) FPKM density distribution for all transcripts. (C) Pearson correlation coefficients for all samples.

protein coding potential using CPC and CNCI. The CPC and CNCI scores in the control and PCV2 infection groups were similar, showing only slight differences (Figs. S1B and S1C). In total, 14,001 non-coding transcripts were predicted by CPC and 13,520 non-coding transcripts were determined using CNCI. Detailed information for the predicted lncRNAs is shown in Table S5. Moreover, lncRNAs were evenly distributed on each chromosome in both the control and PCV2 infection groups (Fig. S2).

## Feature comparison of transcripts

To determine the differences between lncRNAs and mRNAs in IPEC-J2 cells, we compared their transcript structure, sequence conservation, and expression levels. The results showed a contrasting distribution tendency of exon number between mRNAs
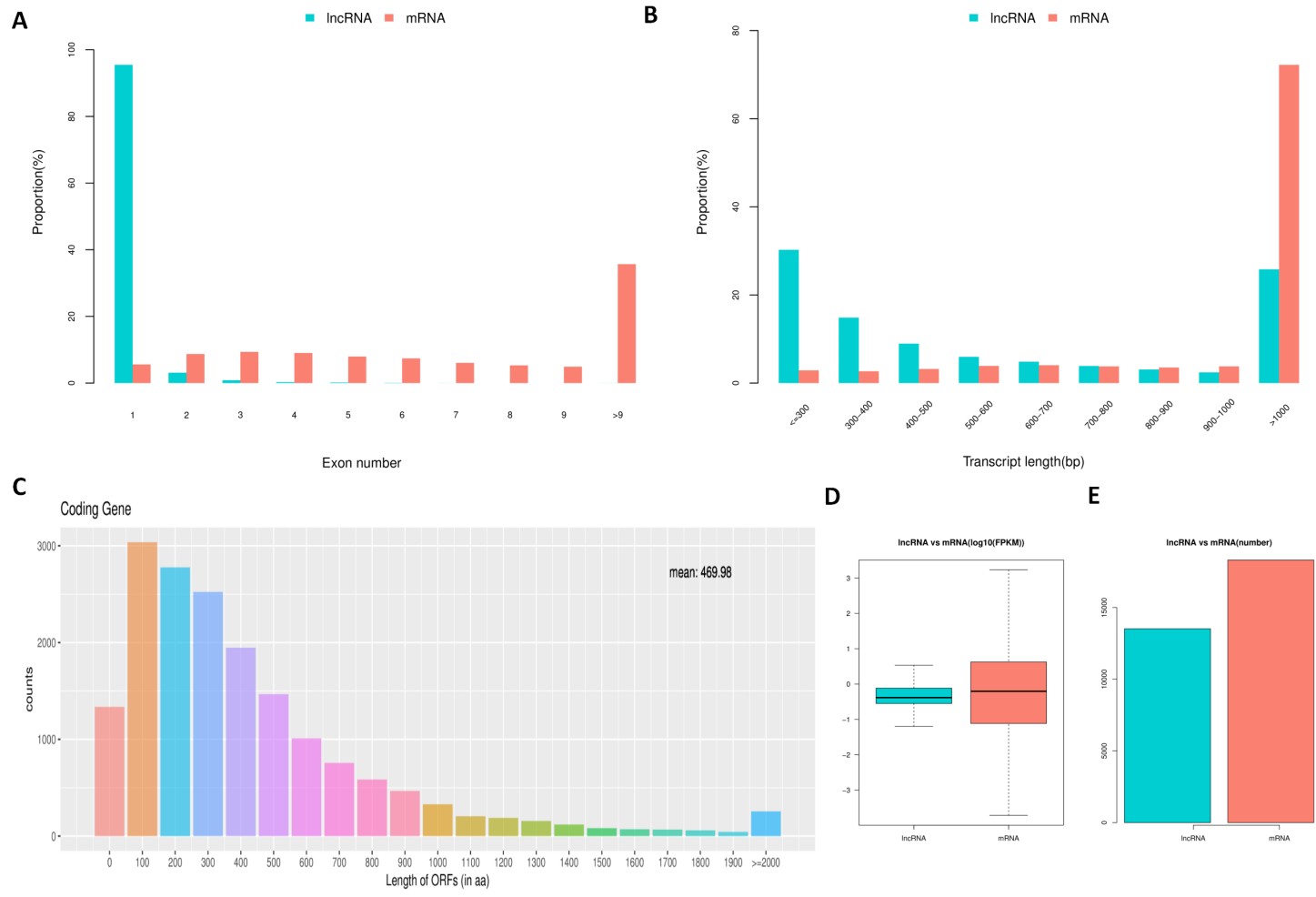

**Figure 4 Comparison between lncRNAs and mRNAs.** LncRNA and mRNA transcripts compared by exon number (A), length (B), ORF length (C), and expression level (D and E).

and lncRNAs (Fig. 4A). The distribution of lncRNAs was enriched on one exon term, which differed from that of mRNAs (Fig. 4A). Additionally, a large proportion of mRNAs were longer than 1,000 base pairs, whereas lncRNAs were generally shorter (Fig. 4B). As expected, most of the lncRNAs contained a comparatively shorter ORF (mean = 61.41) compared to the ORFs in mRNAs (Fig. 4C). Compared to mRNAs, the expression level of lncRNAs was generally lower (Figs. 4D and 4E).

## Characteristics of lncRNA expression levels between the control and PCV2 infection groups

The tentative lncRNAs were quantified by Cuffdiff software using the read count and FPKM analyses. The graphs in Fig. 5A show the lncRNA expression levels in IPEC-J2 cells after PCV2 infection; differentially expressed lncRNAs are also shown. There were 132 up-regulated and 67 down-regulated lncRNAs (P < 0.05) (Fig. 5B; Table S6). The heat map (Fig. 5C) indicates the differentially expressed lncRNAs (P < 0.05) between

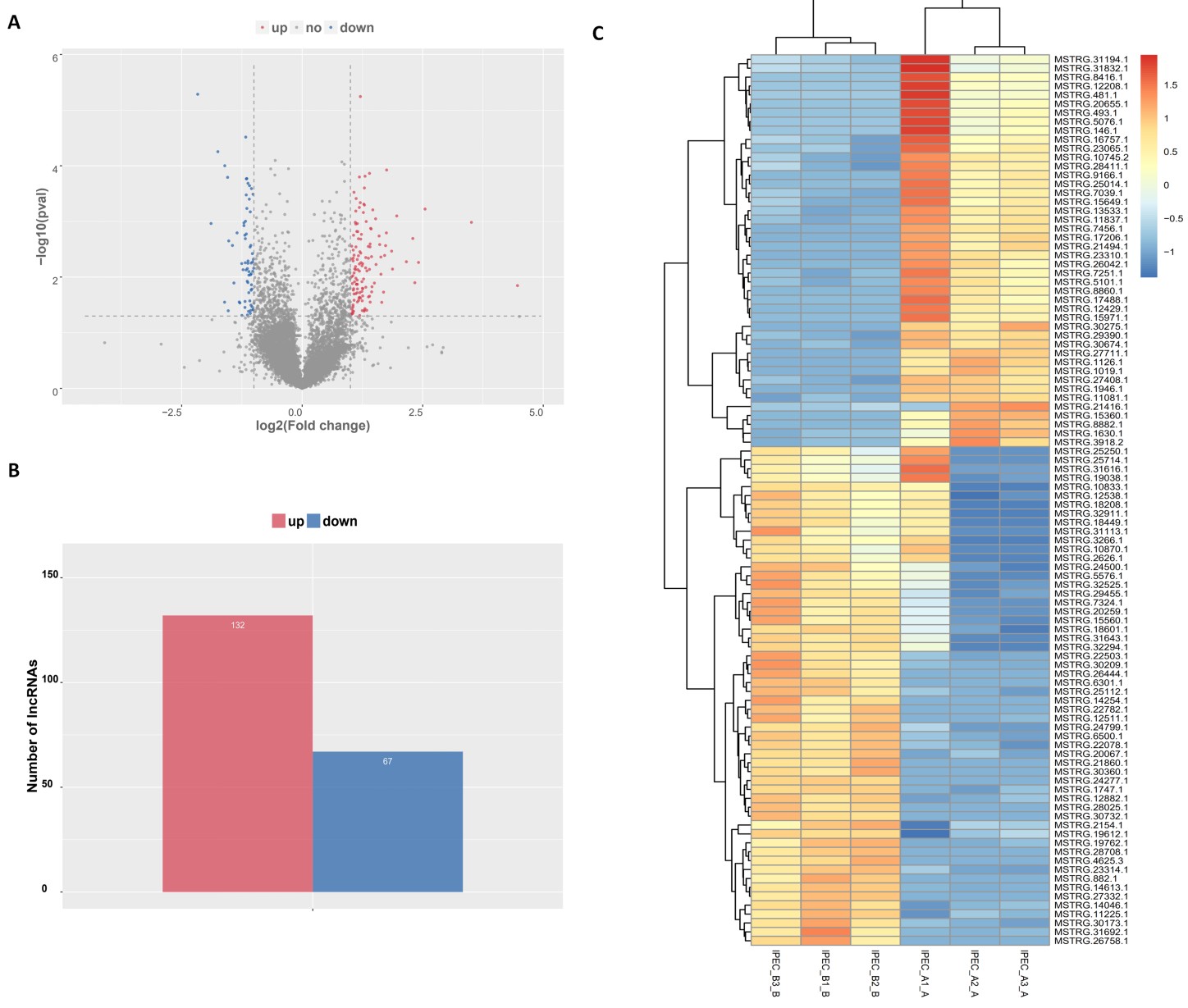

**Figure 5 Characteristics of lncRNA expression levels between PCV2 infection and control groups.** All lncRNA expression levels are shown; differentially expressed lncRNAs are shown in red (up-regulated) or blue (down-regulated) (A). Number of differentially expressed lncRNAs (B). Heat map showing the expressed lncRNAs ($P < 0.05$) in the two groups. Colors from dark blue to orange reveal increasing RNA levels in each group (C).

the control group and the PCV2 infection group. Nine lncRNAs, MSTRG.19762.1, MSTRG.1454.1, MSTRG.641.1, MSTRG.31692.1, MSTRG.385.1, MSTRG.2965.1, MSTRG.15360.1, MSTRG.22503.1, and IMSTRG.5484.1, were selected for validation by qRT-PCR, and the results were consistent with the differential expression observed in the heat map (Figs. 6A–6I). Particularly, MSTRG.31692.1 was down-regulated by 20-fold (Fig. 6I).

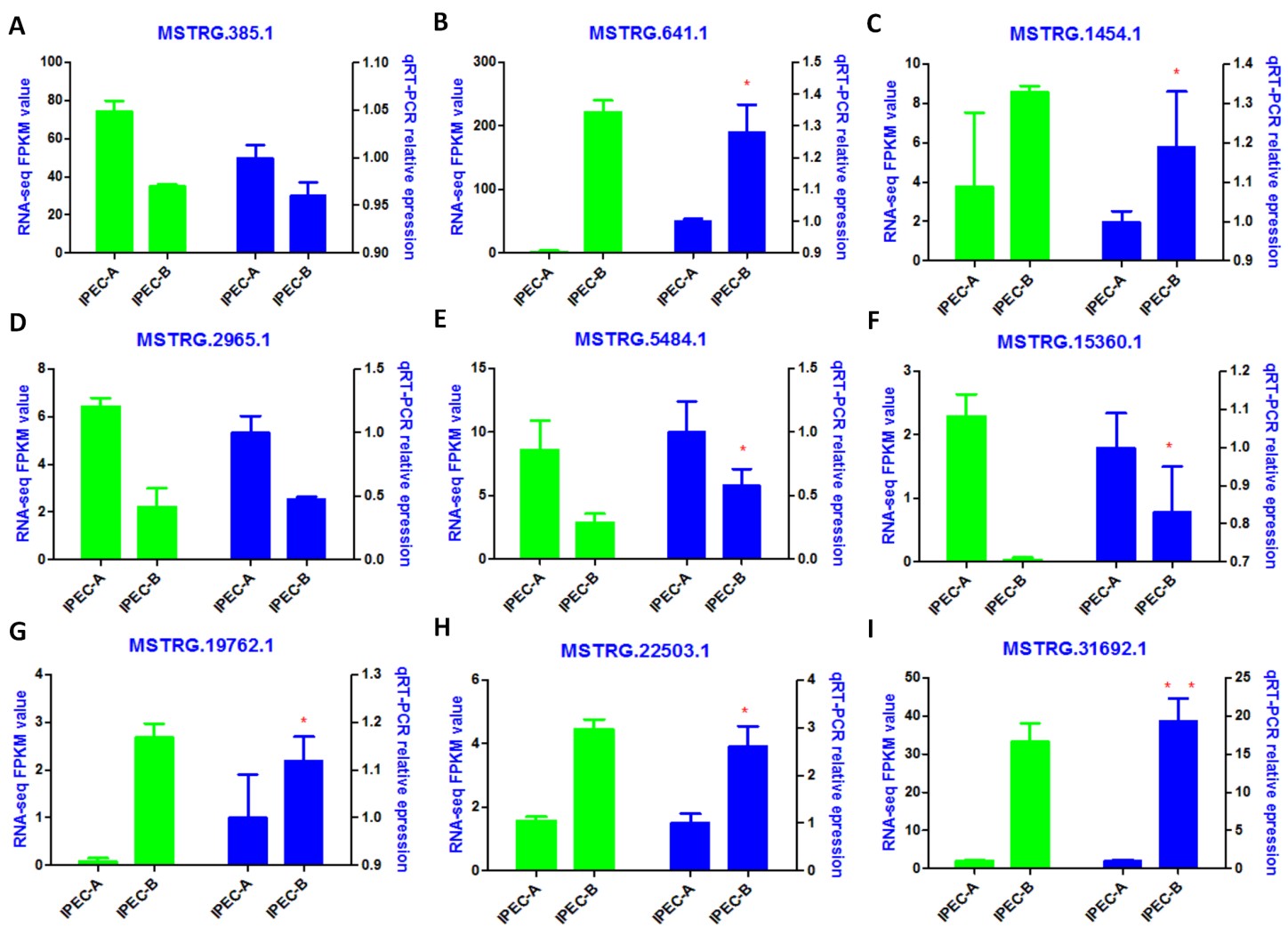

**Figure 6 Validation of RNA-Seq data by qRT-PCR.** Left *Y*-axis shows the FPKM values of the selected lncRNAs (A–I) using RNA-seq, whereas the right *Y*-axis shows the relative expression levels of selected lncRNAs (A–I) using qPCR. * indicates $P < 0.05$.

## Prediction and functional analysis of differentially expressed lncRNA trans-regulated target genes

All reads were assembled into 31,836 transcripts as either mRNAs or lncRNAs. Briefly, there were 18,316 mRNAs, including 623 mRNAs predicted for differentially expressed lncRNA target genes. Of these 623 mRNAs, the expression of 373 mRNAs was altered by >two-fold. Genes can be regulated by lncRNAs in *cis* or in *trans*, and both regulatory mechanisms may play an important role in pathological and biological processes in pigs. Of the 373 differentially expressed genes, 362 were lncRNA trans-regulated target genes. Thus, we focused on differentially expressed lncRNA trans-regulated target genes in this study. The list of differentially expressed lncRNA target genes is shown in Table S7.

To further evaluate the relationship between lncRNAs and target genes, we mapped the nine validated lncRNAs and their target genes into the lncRNA-mRNA regulatory

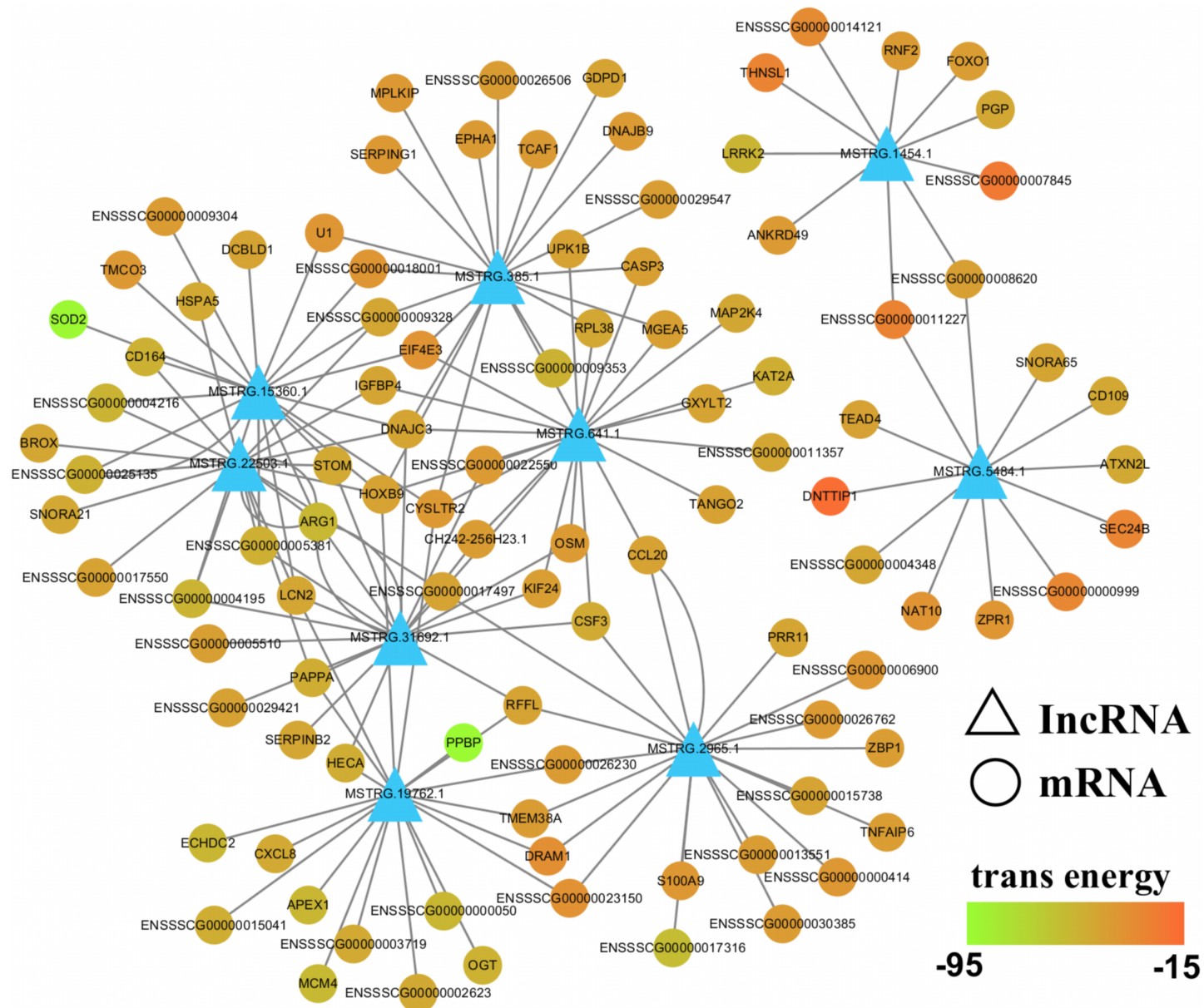

**Figure 7 LncRNA-mRNA regulatory network between validated lncRNAs and target genes.** View of lncRNA-mRNA regulatory network according to nine validated lncRNAs and their target genes.

network (Fig. 7; Table S8). This interaction network was delineated using the Cytoscape software (v3.7.0). From this network, we inferred that lncRNAs may play a central role in PCV2 infection, as they regulate numerous target genes. Additionally, some of these target genes, including *SOD2*, *TNFAIP3*, *ARG1*, *SERPINB2*, *VLDLR*, *HSPA5*, and *LCN2*, are associated with infectious diseases, suggesting that lncRNAs respond to PCV2 infection by regulating these genes.

Next, differentially expressed target genes of the lncRNAs in *trans* were subjected to GO enrichment. The histogram in Fig. 8A shows the number of genes for a term

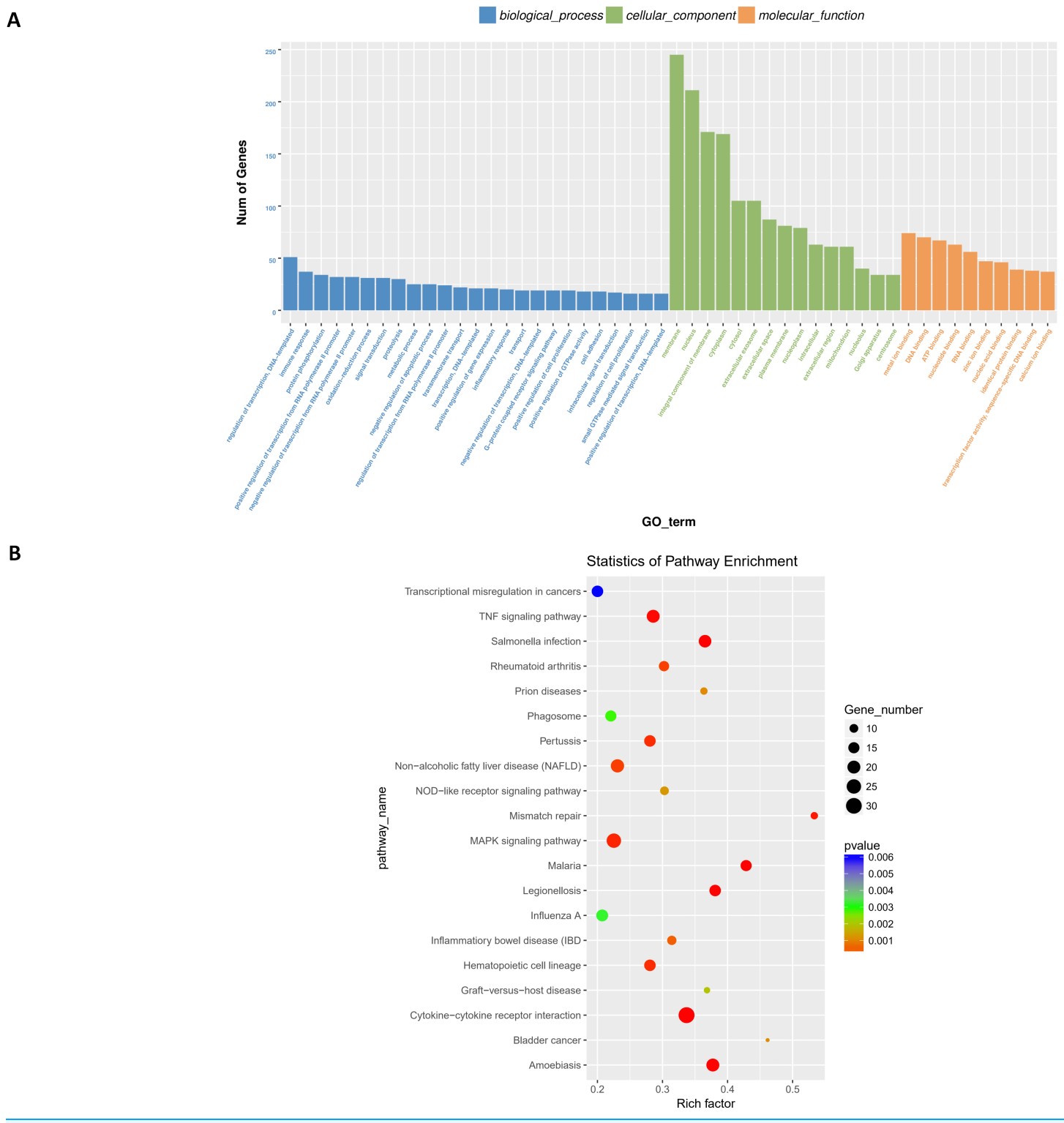

**Figure 8 Functional analysis of target genes regulated by differentially expressed lncRNAs in *trans*.** For *trans* interactions, the GO enrichment histogram (A), GO terms (B), and KEGG pathway enrichment scatter plot (B) are shown.

distributed across biological processes, cellular components, and molecular functions. The histogram also shows the enriched GO terms of molecular function, including metal ion binding, DNA binding, ATP binding, nucleotide binding, RNA binding, zinc ion binding, nucleic acid binding, identical protein binding, transcription factor activity, sequence-specific DNA binding, and calcium ion binding (Fig. 8A); detailed information is shown in Table S9.

Finally, we performed KEGG pathway enrichment for the differentially expressed lncRNA target genes. The 44 critical pathways with low *P*-values ($P < 0.05$) and 501 genes are shown in Fig. 8B. Briefly, most enriched pathways were related to the TNF signaling pathway, *Salmonella* infection, Pertussis, MAPK signaling pathway, cytokine-cytokine receptor interaction, Influenza A, and Amoebiasis (Table S10).

## DISCUSSION

Porcine circovirus type 2 is associated with PMWS and other porcine diseases that have a major negative impact on the global pig industry. In our efforts to characterize non-coding RNAs that may be involved in porcine diseases, we identified 13,520 novel porcine lncRNAs. Similarly, a large number of novel pig lncRNAs was also identified by another group, who predicted 12,867 novel lncRNAs in porcine alveolar macrophages after infection of the HP-PRRSV GSWW15 strain and the North American strain FL-12 (*Zhang et al., 2017b*). However, these numbers are low compared with those in other animals, and further characterization of porcine lncRNAs is required. The lncRNAs identified here share universal characteristics with other mammals including fewer exons, shorter length and lower expression level than protein-coding genes (*Cabili et al., 2011*; *Ravasi et al., 2006*; *Ulitsky & Bartel, 2013*). Thus, although further characterization is required, the present study provides a reference for studying lncRNAs in other species.

There is strong evidence that lncRNAs play a clear role in viral infection. For example, lncRNA negative regulator of antiviral response modulates antiviral responses by suppressing the initiation of interferon-stimulated gene transcription (*Ouyang et al., 2014*). In addition, the upregulation of lncRNA-CMPK2 contributes to the negative regulation of the interferon response (*Kambara et al., 2014*). To date, the roles of lncRNAs in viral-host interactions during PCV2 infection are still unclear. To our knowledge, this is the first report that characterizes the expression profiles of lncRNAs in the non-transformed columnar epithelial cell line IPEC-J2 after PCV2 infection. Post PCV2 infection, we identified 199 differentially expressed lncRNAs, which appear to modify genes associated with viral infection, such as *SOD2*, *TNFAIP3*, *ARG1*, *SERPINB2*, *VLDLR*, *HSPA5*, and *LCN2* (Table S11). For example, *TNFAIP3* is involved in influenza A virus infection (*Maelfait et al., 2012*), whereas *ARG1* suppresses arthritogenic alphavirus infection (*Burrack et al., 2015*) Therefore, PCV2 infection-associated lncRNAs may modulate viral infection through regulating these targeted genes.

The lncRNAs predicted as regulators of genes related to infectious diseases include MSTRG.4625, MSTRG.8436, MSTRG.4146, MSTRG.5886, MSTRG.5870, MSTRG.4146, and MSTRG.4592 (Table S11). However, since every targeted gene is regulated by
several lncRNAs (Table S11), further studies are required to determine the mechanism of combinatorial control.

Our functional enrichment analysis revealed that processes including DNA binding, transcription factor activity and identical protein binding are closely associated with PCV2 infection. Consistent with this, PCV2 induces the activation of transcription factor nuclear factor kappa B (NF-kappa B) by increasing DNA binding activity (Han et al., 2017; Wei et al., 2008). Another study revealed that PCV2 protein ORF4 induces apoptosis by binding to mitochondrial adenine nucleotide translocase 3 (Lin et al., 2018). These data suggest that lncRNAs may respond to PCV2 infection through regulating DNA binding, transcription factor activity and identical protein binding.

The most enriched pathways in PCV2 infection include the TNF signaling pathway, Salmonella infection, Pertussis, MAPK signaling pathway, cytokine-cytokine receptor interaction, Influenza A, and Amoebiasis. Salmonella infection often occurs concurrently with PCV2-associated disease (Takada-Iwao et al., 2011); thus, lncRNAs involved in PCV2 infection may also play roles in Salmonella infection. However, no studies have directly demonstrated the involvement of the other pathways we identified in PCV2 infection. A previous study has demonstrated that PCV2 vaccination may protect piglet against PCV2 infection through inducing TNFα production (Koinig et al., 2015). Notably, these pathways, such as the TNF signaling pathway, MAPK signaling pathway, and cytokine-cytokine receptor interactions, play major roles in the host inflammatory response to numerous infectious diseases (Benedict, Banks & Ware, 2003; Gong et al., 2011; Li et al., 2011; Maegraith & Harinasuta, 1953), suggesting that swine lncRNAs may contribute to host inflammation during PCV2 infection. Further studies should focus on exploring the underlying mechanisms by which swine lncRNAs affect PCV2 infection. PCV2 is associated with PMWS and other porcine diseases that have a major negative impact on the global pig industry. In our efforts to characterize non-coding RNAs that may be involved in porcine diseases, we identified 13,520 novel porcine lncRNAs. Similarly, a large number of novel pig lncRNAs was also identified by another group, who predicted 12,867 novel lncRNAs in porcine alveolar macrophages after infection of the HP-PRRSV GSWW15 strain and the North American strain FL-12 e. However, these numbers are low compared with those in other animals, and further characterization of porcine lncRNAs is required. The lncRNAs identified here share universal characteristics with other mammals including fewer exons, shorter length and lower expression level than protein-coding genese. Thus, although further characterization is required, the present study provides a reference for studying lncRNAs in other species.

## CONCLUSIONS

In summary, the current study reveals candidate lncRNAs associated with PCV2 infection and the cellular signaling pathways that they modulate. These findings could be valuable in designing novel potential strategies to identify the molecular mechanisms underlying PCV2-associated diseases. In turn, this will facilitate the development of antiviral strategies against PCV2 that will benefit animals in the global pig industry.

### Funding

This study was supported by the General Program of the National Natural Science Foundation of China (Grant Nos. 31372406 and 31571432), the Hunan Provincial Natural Science Foundation of China (Grant No. 2018JJ2177), and the Research Foundation of Hunan Provincial Education Department, China (Grant No. 15A086). The funders had no role in study design, data collection and analysis, decision to publish, or preparation of the manuscript.

### Grant Disclosures

The following grant information was disclosed by the authors:
General Program of the National Natural Science Foundation of China:
31372406 and 31571432.
Hunan Provincial Natural Science Foundation of China: 2018JJ2177.
Research Foundation of Hunan Provincial Education Department, China: 15A086.

### Competing Interests

The authors declare that they have no competing interests.

### Author Contributions

- Manxin Fang conceived and designed the experiments, performed the experiments, analyzed the data, prepared figures and/or tables, authored or reviewed drafts of the paper, approved the final draft.
- Yi Yang analyzed the data, prepared figures and/or tables.
- Naidong Wang performed the experiments.
- Aibing Wang performed the experiments.
- Yanfeng He analyzed the data.
- Jiaoshun Wang analyzed the data.
- You Jiang contributed reagents/materials/analysis tools.
- Zhibang Deng conceived and designed the experiments, authored or reviewed drafts of the paper, approved the final draft.

### Data Availability

Data is available at the Sequence Read Archive (SRA) database, accession number PRJNA492887.

### Supplemental Information

Supplemental information for this article can be found online at http://dx.doi.org/10.7717/peerj.6577#supplemental-information.

Peerj

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
