# Peer review of "Genome-wide analysis of long non-coding RNA expression profile in porcine circovirus 2-infected intestinal porcine epithelial cell line by RNA sequencing"

_PeerJ, doi:10.7717/peerj.6577_

## Round 0.1 · original submission · Major Revisions

Please revise the manuscript with special attention to the remarks regarding the details about RNA purity and concentration, as well as for the description of the computational methods and outputs of the analyses performed.

Reviewer 1 ·

Basic reporting

Tha paper seems overall well written, although a minor review in the grammar might be appropriate. The references seem to cover all aspects of the subject. The overal structure of the article is well displayed, as are the figures.
It seems an interesting and quite worthy paper to be published.

Experimental design

I am not qualified to examine the methodology employed in this paper. The overal feeling is that it seems quite good, though the opinion of someone who is more familiar with such methodology would be welcome.

Validity of the findings

The results seem quite interesting and the subject is highly innovative. Data seems robust and open the oportunity for a number of other studies which might use this approach.
I would suggest that this paper be sent to someone more familiar with this field than I.

Additional comments

The results seem quite interesting and the subject is highly innovative. Data seems robust and opens the oportunity for a number of other studies which might use this approach. However, I would suggest that this paper be sent to someone more familiar with this field than I.

Reviewer 2 ·

Basic reporting

no comment

Experimental design

no comment

Validity of the findings

no comment

Additional comments

This manuscript describes genome-wide profiling of long noncoding RNA in porcine epithelial cell lines (normal/infected by PVC2) by RNA-seq; with the goal of providing insights into the biological changes regulated by lncRNAs. Since no one has revealed the lncRNA changes in this aspect, the authors’ results are interesting and might have inspirations to researchers on the studies of porcine circovirus-associated disease (PCVAD). However, there are some clarifications that are required for this manuscript to be suitable for publication. My specific comments are indicated below. There are still numbers of grammatical errors throughout that need to be addressed to provide clarity.

1. Lines 20-21: How “100 million raw reads” was calculated, for each sequencing library or all sequencing libraries. Otherwise, the “100 million” was not representative for your six libraries, and according to TableS1, we could easily fond the largest library contains nearly 150 million raw reads (148.7 million). The author should revise appropriately.
2. Lines 51-52: The provenance or article for the lncRNA definition should be cited clearly.
3. Lines 122-123: How you defined a read was low quality read? Did you allow mismatches when discarding reads which containing adapters? And what the percentage of “N” in your ploy-N sequences filtered?
4. Lines 127-129: What were the parameters of TopHat? Did you use the default settings? If not, the parameters should be given. And the genome (suscrofa 10.2) used while mapping should be stated here—the first time it appears or used in your article. I also wondered why you chose suscrofa 10.2 instead of suscrofa 11.1. Otherwise you should cite the article of TopHat.
5. Lines 131-133: Again, the parameters were not provided for StringTie and Cufflinks as well as the version of StringTie. And to my opinion, the articles you cited here were not proper. For example, in Shao & Kingsford 2017 they introduced an assembly software (Scallop) and compared the accurate while applied to assembly work with other softwares.
6. Lines 135-140: The versions and parameters of CPC and CNCI should be provided as well as citing proper articles. Moreover, I wonder the steps before coding potential analysis. So, I think it’s better to described a lncRNA identification method rather than coding potential analysis.
7. Line 143: How to classify you lncRNAs into three types (lincRNA, intronic lncRNA and anti-sense lncRNA)? And the definition of different types should be defined.
8. Lines 149-153: Did you only keep coding genes between 10-100K upstream or downstream a lncRNA? Why don’t you consider the genes within 10K? How did you identify trans interacted genes with expression level? Pearson correlations or Spearman correlations? The method should be stated in detail.
9. Lines 155-158: Parameters should be provided either.
10. Lines 165-166: I guess the author misused the words of “less than”.
11. Lines 172 and Table 1: The word “Products length” is much better for the fourth column of Table 1.
12. Lines 178-182: Please checking how to cite an R package. And please find and cite the correct article for KOBAS.
13. Lines 227-228: Again, how did the number of three different types calculated?
14. Line 235 and Fig.S2: Figures’ legend should be described clearly. Take Fig.S2 for example, we couldn’t know the colors of different circles stand for. Even, we don’t know the units for the numbers beside each chromosome, single base or kilobase? Other figures should be checked either.
15. Lines 252-253: Please clarify the cluster method of the heatmap, what’s the color bar stands for? Were the FPMK value transformed by a formula?
16. Lines 265-266: Of the 373 DEGs, 362 were lncRNA trans-regulated target genes. Did you mean these 362 DEGs have related lncRNAs within 10-100Kb upstream or downstream?
17. Lines 270-271: The lncRNA-mRNA regulatory network was shown in Fig 6, not Fig 5. Please illustrate the version for Cytoscape software.
18. Lines 276-287: This paragraph was described according Fig 5. Low P values and large numbers of genes were un-clarified, the P value and gene number should be demonstrated clearly.
19. For supplementary tables, suitable tittles with few words or annotations in it that are better for the readers’ understanding.
20. Sometimes, the authors' discussion appears to be a repeat of the data presented and enrichment analysis in the results section such as lines 323-327 and lines 338-341. It is suggested that the authors provide an expansion of detail in the discussion section as well as a comparison to other studies.
21. The nine lncRNAs in lines 253-255 were picked for qRT-PCR validation and lncRNA-mRNA regulatory network construction, and even the author found MSTRG.31692.1 was 20-fold-changes down-regulated. Are there one or more genes regulated by MSTRG.31692.1 or others? Whether the regulated genes were reported previously? If so, the correlations and locations should be provided.

·

Basic reporting

Professional English used in the present study.
The literature references is fit.
The article structure, figs, and tables are fit.

Experimental design

The experimental design is fit.

Validity of the findings

This study provides new insight into the mechanism underlying of PCVAD.
Conclusion are well stated.

Additional comments

1. Line 113-118: The results from NanoDrop ND-1000 and Agilent 2100 Bioanalyzer should be added in the present study. The RNA amount and purity are very important to construct the cDNA libraries. In addition, I suggest that the preparation for library and sequencing should be listed in this section, but using the references. Then, authors should add the company name which provided the service for constructing cDNA libraries and sequencing.
2. Line 128: Authors should add the version of the porcine reference genome in the present study.
3. In the Materials and Methods section, it is not so clear how the authors screened and identified the putative lncRNAs. Then, It would be helpful if they illustrate a figure to clarify this process as DOI:10.1095/biolreprod.115.136911 and DOI: 10.1016/j.ygeno.2017.07.001.
4. Line 256-257: Fig. S3 should be showed in the main manuscript but in the additional files.
5. English must be edited again. There are some grammatical errors in the manuscript. Such as line 97 CO2, Line 102 mL, Line 345 differentially, etc.
6. In the present study, 13,520 lncRNAs were identified. I suggest that authors predicted the protein coding potential using CPC, CNCI, Pfamscan, and phyloCSF. It will decrease the false positive in identifying novel lncRNAs using four software in this section.
7. 13,520 lncRNAs were identified in the present study. How many pig known lncRNA were identified? How many pig novel lncRNA were identified?

---

## Round 0.2 · Minor Revisions

The authors should include more detail about the library types used and efforts are still need regarding typo and grammar, as noted by on the the reviewers. Some corrections are also needed in the figures.

Reviewer 2 ·

Basic reporting

no comment

Experimental design

no comment

Validity of the findings

no comment

Additional comments

In the revised manuscript the authors have addressed many, but there are still some issues that should be addressed.
1. Although, the authors have provided parameters of almost all software’s and illustrated that “the preparation for library and sequencing were performed according to previous studies (Wang et al. 2016)”, still, the library types (FR, RF, PE or SE etc.) should be added for better understanding of the data analysis.
2. Throughout the tracked changes manuscripts, there are still many grammatical errors, a modification by some one of English-native could improve the paper for clearly understanding.
3. The author only provided an expansion of lncRNA characters-comparison, but we readers prefer lncRNA researches on PCV2. Otherwise, the repeat parts that I mentioned in my last review are not revised.
4. In the fig.5C, please clarify what’s the value of color bar stands for and what’s the method of Hcluster heatmap? If the author did Hcluster heatmap with log2-transformed FPKM value?
4. Figures and legends should be checked and revised again, for example, the words “Fig.3” were in the topleft of “peerj-31169-FigS2_Distribution_of_lncRNAs_along_each_chromosome.png”.
5. The tables in the supplemental files seems to be processed files, many redundant information was un-necessary. For example, the class code, chromosome, start and end were repeated in several tables.
6. In the Figure.1, the author filtered transcripts with CPC-scroe < 0, but in the method part the author wrote “All transcripts with CPC scores ≤-1 were removed”. Please clarify it.
7. For the softwares, R packages, not only the versions were needed but also the cited papers.
8. In the discussion part, the author explained too much of using PCV2-infected IPEC-J2 cells in this study. It should be stated simplified and described in the introduction part.

·

Basic reporting

In my review, the professional English used throughout in the manuscript. The manuscript included sufficient introduction and background to demonstrate the long non-coding RNA
expression profile in porcine circovirus 2-infected intestinal porcine epithelial cell line. The structure of the article is conformtable to an acceptable format of ‘standard sections’. In my opinion, this manuscript is very fit to publish in Peer J.

Experimental design

The research question in the manuscript is relavant and meaningful. The research have been conducted in conformation with the prevailing ethical standards in the field. In addition, The methods in the manuscript were described with suffient information.

Validity of the findings

The manuscript reported that 13,520 lncRNAs were identified from intestinal porcine epithelial cell line after infection of Porcine circovirus 2. These results provides new insight into the
mechanism underlying of PCVAD. The RNA-seq data and other data can support the conclusions in the manuscript, and the conclusions were appropriately stated.

Additional comments

Thank you to the authors for making the requested corrections and I think the manuscript is well-organized and fit for publication.

---

## Round 0.3 · accepted · Accept

The main criticisms of the referees were answered by the authors.

#